# Non-Destructive Detection of Golden Passion Fruit Quality Based on Dielectric Characteristics

**Fan Lin, Dengjie Chen, Cheng Liu and Jincheng He ***

College of Mechanical and Electrical Engineering, Fujian A&F University, Fuzhou 350001, China; 1211298004@fafu.edu.cn (F.L.); chen_dj@fafu.edu.cn (D.C.); 1211298010@fafu.edu.cn (C.L.)
* Correspondence: jhe@fafu.edu.cn

**Abstract:** This study pioneered a non-destructive testing approach to evaluating the physicochemical properties of golden passion fruit by developing a platform to analyze the fruit's electrical characteristics. By using dielectric properties, the method accurately predicted the soluble solids content (*SSC*), *Acidity* and pulp percentage (*PP*) in passion fruit. The investigation entailed measuring the relative dielectric constant ($\varepsilon'$) and dielectric loss factor ($\varepsilon''$) for 192 samples across a spectrum of 34 frequencies from 0.05 to 100 kHz. The analysis revealed that with increasing frequency and fruit maturity, both $\varepsilon'$ and $\varepsilon''$ showed a declining trend. Moreover, there was a discernible correlation between the fruit's physicochemical indicators and dielectric properties. In refining the dataset, 12 outliers were removed using the Local Outlier Factor (LOF) algorithm. The study employed various advanced feature extraction techniques, including Recursive Feature Elimination with Cross-Validation (RFECV), Permutation Importance based on Random Forest Regression (PI-RF), Permutation Importance based on Linear Regression (PI-LR) and Genetic Algorithm (GA). All the variables and the selected variables after screening were used as inputs to build Extreme Gradient Boosting (XGBoost) and Categorical Boosting (Cat-Boost) models to predict the *SSC*, *Acidity* and *PP* in passion fruit. The results indicate that the PI-RF-XGBoost model demonstrated superior performance in predicting both the *SSC* ($R^2 = 0.9240$, RMSE = 0.2595) and the *PP* ($R^2 = 0.9092$, RMSE = 0.0014) of passion fruit. Meanwhile, the GA-CatBoost model exhibited the best performance in predicting *Acidity* ($R^2 = 0.9471$, RMSE = 0.1237). In addition, for the well-performing algorithms, the selected features are mainly concentrated within the frequency range of 0.05–6 kHz, which is consistent with the frequency range highly correlated with the dielectric properties and quality indicators. It is feasible to predict the quality indicators of fruit by detecting their low-frequency dielectric properties. This research offers significant insights and a valuable reference for non-destructive testing methods in assessing the quality of golden passion fruit.

**Keywords:** golden passion fruit; dielectric properties; feature screening; model establishment; non-destructive testing

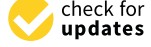



## 1. Introduction

Passion fruit is a perennial climbing vine plant belonging to the Passifloraceae family and Passiflora genus. There are approximately 50–60 edible species of passion fruit found worldwide. They are highly cherished for their unique aroma and nutritional value [1,2]. Among them, yellow passion fruit accounts for about 95% of the world's commercial Passiflora output [3,4]. As a distinctive fruit in southern China, the yellow passion fruit industry has rapidly developed in recent years and holds considerable growth potential. However, passion fruit is primarily harvested manually, leading to issues like inconsistent quality and mixing of ripe and unripe fruit [5]. With the growing demand for passion fruit in both domestic and international markets [6], manual quality assessment hinders the standardized and large-scale development of passion fruit products, making it challenging

to gain a competitive edge in the market. Therefore, establishing a non-destructive quality testing method for passion fruit is essential to meet the needs of the industry's development.

Many scholars, both domestically and internationally, have conducted research on the quality assessment of passion fruit. The primary detection technologies include spectroscopy [7], the electronic nose [8], GC-MS [9,10], the Mechanical Property Test [5,11] and others. Most of these methods require destructive testing. However, Maniwara et al. utilized a non-destructive method using visible and short-wave near-infrared spectroscopy through interacting and transmission measurements to evaluate the physicochemical quality of ripe passion fruit. It is important to note that passion fruit of different maturities exhibit significant variations in their physicochemical indicators, and those with lower maturity tend to have noticeably thicker peels. The peel thickness and internal seeds can significantly impact the efficiency of spectroscopic non-destructive testing methods. Compared to these techniques, non-destructive testing based on dielectric properties is more efficient and reliable and involves lower equipment costs.

Fruit undergoes a series of changes during the ripening process, both at the macro- and microscopic levels. Macroscopically, these changes are observed in terms of hardness, soluble solids content, acidity, pulp percentage and color. From a microscopic perspective, there is a formation of a bioelectric field generated by numerous charged particles within the fruit. As substances undergo transformation during the growth and ripening process, the amount of charge carried by various chemical substances within the fruit's internal tissues, as well as the spatial distribution of charges, may also change. These microscopic changes in the electric field have an impact on the fruit's dielectric properties at the macroscopic level [12,13].

The study of the relationship between dielectric properties and the quality of fruit, as well as the exploration of methods for the non-destructive evaluation of fruit quality based on dielectric properties, has attracted widespread attention. Both domestic and international research on fruit and vegetables like bananas [14], kiwifruit [15], carrots [16], Korla pears [17,18] and mangoes [19] involved measuring their electrical properties at low frequencies to determine the physicochemical qualities. The current research methods have shifted from early statistical analysis of dielectric properties to establishing multivariate models based on dielectric properties combined with various algorithms. Fazayeli et al. [15] developed a model using an Artificial Neural Network with dielectric property features as input to predict the hardness, soluble solids content and pH value of kiwifruit, achieving $R^2$ values of 0.92, 0.91 and 0.86, respectively. Lan et al. [17] used equivalent parallel capacitance, quality factor, loss factor, equivalent parallel resistance, complex impedance and equivalent parallel inductance as model inputs. And compared the predictive performance of three models(GRNN, BPNN and ANFIS), in forecasting the soluble solids content of Korla pear. Ibba et al. [20] measured the bioimpedance data of strawberries at both mature and immature stages, between frequencies of 20–300 kHz. They utilized six of the most commonly used supervised machine learning classification techniques to evaluate their effectiveness in predicting strawberry fruit ripeness. Extensive research indicates that machine learning algorithms have been widely applied in predicting and classifying fruit and vegetable quality. Appropriate postharvest treatment methods for fruit and vegetables of different maturity can ensure quality during the shelf life and increase economic value [21].

This paper focuses on "Bale Huangjinguo" passion fruit [22] as the research subject, analyzing the trends in dielectric properties under different frequencies and growth degree days. The relative dielectric constant and dielectric loss factor at 34 frequencies within the low-frequency range of 0.05–100 kHz are employed as feature representations. Anomaly values are filtered out using the LOF algorithm, followed by the application of four diverse feature selection methods: RFECV, PI-RF, PI-LR and GA. Ultimately, XGBoost and CatBoost models were established to predict the quality indicators of passion fruit, such as *SSC*, *Acidity* and *PP*, and we compared the predictive performance under various feature selection methods and different models. This study demonstrates the feasibility of combining

low-frequency dielectric properties and machine learning for non-destructive testing of passion fruit quality.

## 2. Materials and Methods

### 2.1. Test Samples

From July to October 2021, a total of 192 experimental samples were collected in three batches from fruit orchards in Pinghe County, Zhangzhou City, Fujian Province. The samples had growth degree days ranging from 700 to 1400 °C·d. The variety selected for the passion fruit was "Bale Huangjinguo", with an average sample weight of 86.7 ± 10.15 g, horizontal diameter of 63.39 ± 3.26 mm and longitudinal diameter of 64.04 ± 2.46 mm. The experimental samples had full fruit shape and showed no obvious external defects or damage from pests and diseases, as shown in Figure 1. All the samples were cleaned, air-dried and individually numbered. The samples' dielectric properties and physicochemical indices were measured in the laboratory at a room temperature of 25 ± 1 °C on the same day.

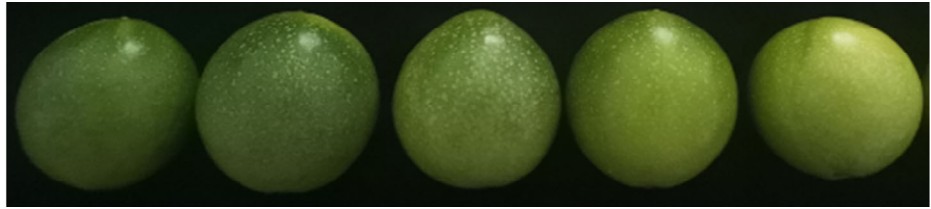

**Figure 1.** Passion fruit sample pictures; these samples become more mature from left to right.

### 2.2. Measurement Equipment and Methods

#### 2.2.1. Dielectric Parameter Measurement

The passion fruit dielectric properties acquisition system, as shown in Figure 2, primarily consists of an LCR digital electric bridge tester (TH2830 type, Tonghui electronics equipment company, Changzhou, China), a fruit electrical parameter testing platform (self-made) and a computer. The measurement method involves the placement of the sample between two parallel plates in direct contact with them to assess the sample's dielectric properties, which encompass $\varepsilon'$ and $\varepsilon''$. The entire system is housed within a sealed shielded enclosure to safeguard against external interference.

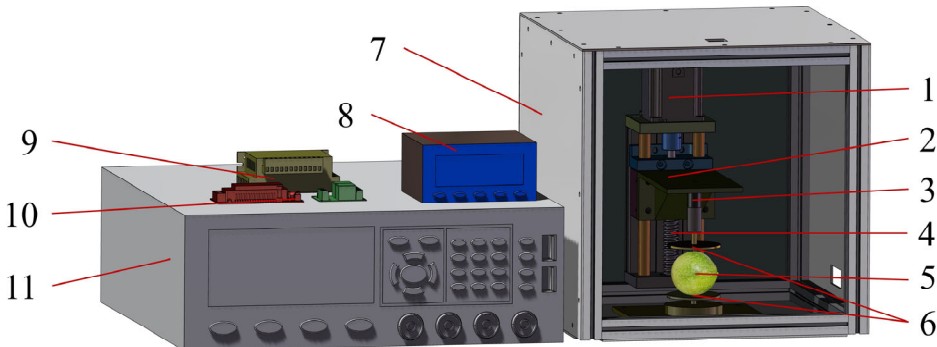

**Figure 2.** Electrical characteristics testing platform. Among them are 1. Electrical machinery, 2. Support stand, 3. Force sensor, 4. Lead screw, 5. Sample, 6. Electrodes, 7. Shielding box, 8. Force display, 9. Motor drive module, 10. Microcontroller, 11. LCR tester.

The LCR tester has a frequency range of 0.05–100 kHz and a measurement accuracy of 0.05%. Two measurement probes of the LCR tester are connected to the parallel plate capacitor formed by the upper and lower brass plates. Consistency in applying external force during different experimental processes is ensured through a pressure sensor. When the force reaches 1 N, the motor stops running and the dielectric parameters are measured. Data are acquired using the testing system software (TH2832LCR 2.2.21). Samples are

placed horizontally along the longitudinal axis, and data are collected every 120° rotation along the longitudinal axis. The obtained dielectric parameters from three measurements are averaged to represent the dielectric characteristics of the sample.

After placing the test sample between the parallel plates, the equivalent circuit consists of a parallel combination of capacitance *Cp* and conductance *G*. The capacitance value *Cp* and conductance value *G* at the characteristic frequency are measured using an LCR tester. According to Formula (1), the real and imaginary parts of the dielectric constant are calculated as the relative permittivity $\varepsilon'$ and the dielectric loss factor $\varepsilon''$ [23,24], as shown in Formula (2).

$$Z = G + j\omega C_p = j\omega C_0 \left( \frac{C_p}{C_0} - j\frac{G}{\omega C_0} \right) \tag{1}$$

$$\begin{cases} \varepsilon' = \frac{C_p d}{\varepsilon_0 S} \\ \varepsilon'' = \frac{G}{\omega C_0} \end{cases} \tag{2}$$

where $Z$ is the complex impedance, $\Omega$; $C_p$ is the parallel capacitance of the golden passion fruit, F; $d$ is the distance between plates during the test, m; $C_0$ is the vacuum capacitance when the dielectric is air, F; $\varepsilon_0$ is the dielectric constant of free space, taken as $8.85 \times 10^{-12}$ F/m; $S$ is the area of the plates, m$^2$; $G$ is the conductance value, $S$; and $\omega = 2\pi f$; $f$ is the test frequency, Hz.

### 2.2.2. Measurement of Physicochemical Index Parameters

In accordance with the provisions of the current national standard for passion fruit quality grading (GB/T 40748-2021) regarding the physicochemical requirements for passion fruit, this study determined the physicochemical characteristics of the samples, including *PP*, *SSC* and *Acidity*. A total of 192 fruit samples were analyzed, with the following ranges observed: *PP* ranged from 44.1% to 61.8%, *SSC* ranged from 9.9% to 23.8% and *Acidity* ranged from 1.19% to 6.46%.

The following describes the *PP* measurement process: Using an electronic balance, weigh the whole passion fruit. Then, cut the passion fruit in half along its longitudinal axis. Gently scoop out the pulp, thoroughly clean the outer shell of the fruit and then weigh the passion fruit peel. Record measurements in grams (g).

$$PP = \frac{W - W_p}{W} \tag{3}$$

where $W$ is the total fruit weight and $W_p$ is the peel weight from each fruit.

The following describes the *SSC* measurement process: A digital refractometer/acid meter (Model PAL-BX/ACID-F5, ATAGO, Tokyo, Japan) is used. To obtain the juice for SSC measurement, the passion fruit is manually pressed and filtered through two layers of cotton-linen cloth. At least 3 mL of this extracted passion fruit juice is placed in the measurement area of the instrument, and then the *SSC* value of the juice is measured. This process is repeated three times for each sample and then the average value is taken.

The following describes the *Acidity* measurement process: Using a digital refractometer/acid meter (Model PAL-BX/ACID-F5, ATAGO, Japan), to perform the acid measurement, after diluting 1 g of juice 50 times, at least 3 mL of the diluted solution is placed in the measurement area of the instrument to measure the acid value of the juice. This process is repeated three times for each sample and then the average value is taken. The effectiveness of the digital acid measurement has been confirmed in the research conducted by Rivera et al. [25], as referenced in their study.

### 2.3. Data Analysis and Model Building

### 2.3.1. Definition of Fruit Ripeness

Growth degree days can reflect the comprehensive impact of climate conditions on crop growth and analyze the heat conditions for crops. They can be used to determine the growth stages and corresponding physiological characteristics of crops under certain climate

conditions, thus guiding the timely harvesting of crops [26,27]. This study selects growth degree days (*GDD*) during the maturation process of passion fruit as the fundamental reference for fruit ripeness. The formula for calculating *GDD* for the fruit is as follows [28]:

$$GDD = N * (T - T_L) \tag{4}$$

where GDD are growth degree days, °C·d; *N* is the development period in days; *T* is the daily average temperature in °C; $T_L$ is the lower limit temperature in °C, set at 10 °C.

Observations and statistics in orchards reveal that fruit with *GDD* less than 700 °C·d, due to their insufficient growth period, are still in the fruit expansion period and unlikely to be mistakenly harvested. Fruit are considered fully mature when they naturally fall off the vine after reaching a maximum of 1400 °C·d in *GDD*. Thus, this paper primarily investigates fruit with *GDD* in the range of 700~1400 °C·d. The *GDD* data are sourced from meteorological statistics in Zhangzhou, Fujian, where the fruit are grown. The calculation of *GDD* for each passion fruit starts from the fruit setting date.

### 2.3.2. Modeling Methods

The models utilize dielectric parameters from a specified frequency range as input variables. The LOF algorithm is employed to eliminate anomalies in both the physico-chemical indicators and dielectric characteristics. A variety of feature selection methods, including RFECV, PI-RF, PI-LR and GA, are applied to sift through the dielectric parameters for relevant feature variables. We built XGBoost and CatBoost prediction models separately incorporating the feature selection algorithms. The process diagram is shown in Figure 3. The primary model evaluation metric is the coefficient of determination ($R^2$), with root mean square error (RMSE) as a secondary metric, to compare and analyze the performance of each model combination.

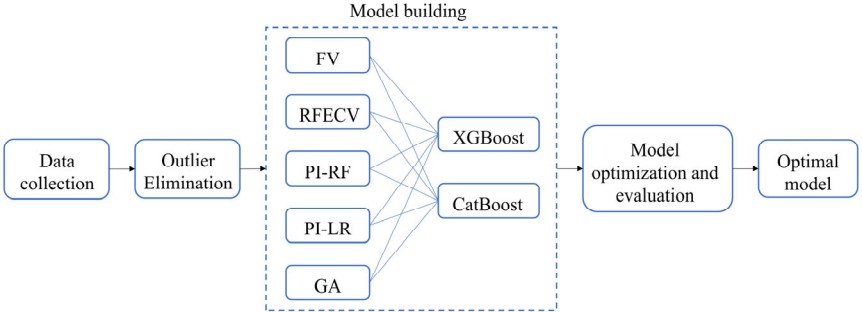

**Figure 3.** Modeling method flow diagram. Through the above process, prediction models for *SSC*, *Acidity* and *PP* are established, respectively.

### 2.3.3. Dataset Splitting

In this experiment, the dataset was divided using the train_test_split() function from sklearn, with a ratio of 9:1 for the training set and the test set. A 10-fold cross-validation method was employed to evaluate the performance of each model, and the results were presented as the mean of each fold. Cross-validation is effective in reducing overfitting, improving generalization and extracting as much useful information as possible from a limited dataset [29].

### 2.3.4. Outlier Elimination

In this experiment, the Local Outlier Factor [30] is used to clean the data by removing invalid and outlier data. This method is based on density-based outlier detection, where the local group size is set to 20 and the distance calculation method is euclidean_distance. It involves calculating the LOF for each point in the dataset to determine how close this value is to 1. In this experiment, outliers were identified and removed separately from the physicochemical indices and dielectric property parameters. If the LOF value is significantly

greater than 1, it is determined to be an outlier. The results from both sets were combined, resulting in the removal of 12 anomalous samples.

### 2.3.5. Feature Extraction Method

The RFECV technique is as follows: Utilize the Cross-Validation Recursive Feature Elimination algorithm for feature selection. Establish a linear model using all features, rate each feature for its importance and eliminate the weakest feature to obtain a subset of important features. Combine Cross-Validation (CV) with the RFE method to score different feature subsets and select the best feature subset, completing the feature selection process [31].

The permutation importance technique is as follows: The algorithm combines Random Forest Regression and Linear Regression to fit the dataset separately, initially determining the prediction accuracy for the original data. Then, the algorithm randomly shuffles the values of a feature while keeping the other characteristics unchanged. After this shuffling, the change in prediction accuracy is compared. The more significant the decrease in accuracy, the higher the dependency of the model's prediction on that particular feature. This process is repeated to rank the features by their importance, thus facilitating feature selection [32].

The Genetic Algorithm technique is as follows: The Genetic Algorithm (GA) first randomly generates initial solutions and evaluates the fitness of each initial solution, transforming the optimization objective of the problem into a fitness function. Then, it undergoes selection, crossover, mutation, replacement and iteration operations based on the theory of biological evolution, continuing until the maximum number of iterations is reached, thereby obtaining the final result for feature selection [33].

### 2.3.6. Machine Learning Algorithm

XGBoost [34] and CatBoost [35] are two novel and popular improved algorithms based on the gradient boosting tree framework. They show significant potential in fields such as biology and medicine. Both algorithms build robust learners by generating and iterating multiple weak estimators to fit the residuals of the final tree model. The XGBoost algorithm incorporates L1 and L2 regularization, effectively controlling model complexity and preventing overfitting. It also has an optimized and unique data structure, resulting in efficient training and prediction speed [36,37]. The CatBoost algorithm is based on symmetric decision trees, can rapidly and accurately make predictions with fewer parameters and is less prone to overfitting. It is particularly efficient and reasonable in handling categorical features [38,39].

## 3. Results and Discussion

### 3.1. Analysis of Dielectric Properties under Different GDD

Figure 4 shows the dielectric parameters of golden passion fruit under different *GDD*. With the increase in the frequency of the test signal, the dielectric parameters $\varepsilon'$ and $\varepsilon''$ of the fruit decrease under different *GDD*. This trend was most significant at low frequencies (0.05 to 10 kHz), a pattern also observed in studies of the dielectric properties of other fruit and vegetables such as tomatoes and mangoes [40,41]. Additionally, at the same frequency, the $\varepsilon'$ and $\varepsilon''$ values were generally lower for ripe samples compared to unripe ones. The main reason for this is that as passion fruit ripens, the membrane permeability and free water content increase. As a result, the ability of the cell membrane in mature fruit to bind charges weakens, leading to a gradual decrease in $\varepsilon'$ and $\varepsilon''$ [42,43]. Overall, in the frequency range of 0.05–5 kHz, there are significant differences in the dielectric properties of passion fruit under different *GDD*.

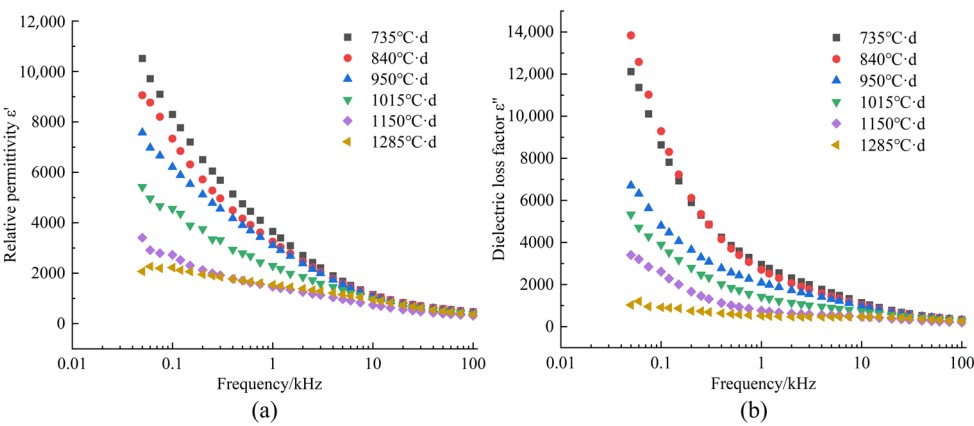

**Figure 4.** Influence of frequency on the dielectric properties of passion fruit: (**a**) relative dielectric constant $\varepsilon'$ and (**b**) dielectric loss factor $\varepsilon''$.

### 3.2. Analysis of the Correlation between Dielectric Properties and Physicochemical Indicators

Pearson correlation analysis between the dielectric parameters and physicochemical indicators at different frequencies is shown in Figure 5. There is a significant correlation between the dielectric parameters ($\varepsilon'$ and $\varepsilon''$) of the summer golden passion fruit and various physicochemical indicators. The dielectric parameters $\varepsilon'$ and $\varepsilon''$ are negatively correlated with *SSC* and *PP*, while they are positively correlated with *Acidity*. The frequency range where $\varepsilon'$ exhibits a higher correlation with the physicochemical indicators is concentrated between 0.05 and 5 kHz, while the frequency range where $\varepsilon''$ shows a higher correlation with the physicochemical indicators is concentrated between 1 and 10 kHz. The correlation coefficients are all around 0.8 within this frequency range.

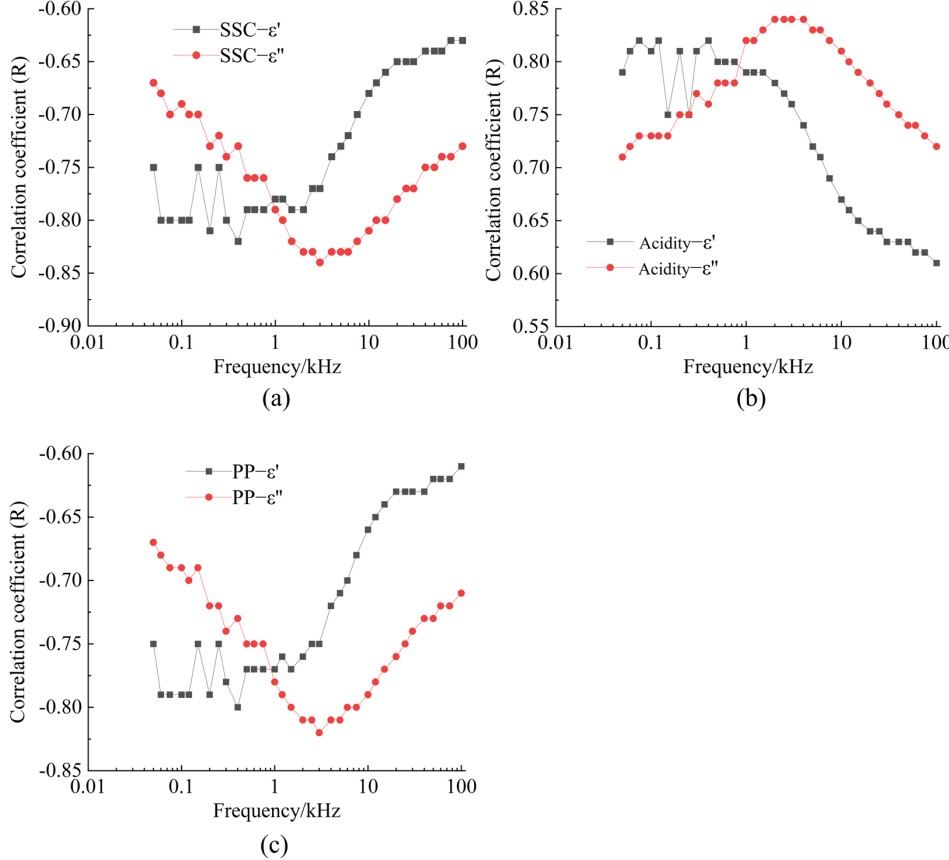

**Figure 5.** Correlation analysis of dielectric parameters and physicochemical properties at different frequencies: (**a**) *SSC*; (**b**) *Acidity*; (**c**) *PP*.

From Figure 4, the dielectric parameters of fruit with different *GDD* show good discrimination within the frequency range of 0.05–5 kHz. Moreover, as indicated in Figure 5, there is a high correlation between the dielectric parameters and physicochemical indicators within the frequency range of 0.05–10 kHz. Based on the above analysis, it can be concluded that selecting feature frequencies within the range of 0.05–10 kHz may be more appropriate. However, overall, the absolute values of the correlation coefficients between $\varepsilon'$ and $\varepsilon''$ at different frequencies and physicochemical indicators are all between 0.6 and 0.85, indicating that the correlations are not very strong. This suggests that it is challenging to accurately predict the physicochemical indicators of golden passion fruit using dielectric parameters at a single frequency. Therefore, to better predict the physicochemical indicators of golden passion fruit using dielectric technology, modeling and analysis should be performed at multiple characteristic frequencies.

### 3.3. Feature Extraction

After removing 12 outlier samples using the LOF algorithm, a physicochemical indicator prediction model was established using 180 passion fruit samples of different ripeness. A total of 68 features were used as inputs, including $\varepsilon'$ and $\varepsilon''$ at 34 different frequency points within the range of 0.05 to 100 kHz (0.05, 0.06, 0.075, 0.1, 0.12, 0.15, 0.2, 0.25, 0.3, 0.4, 0.5, 0.6, 0.75, 1, 1.2, 1.5, 2, 2.5, 3, 4, 5, 6, 7.5, 10, 12, 15, 20, 25, 30, 40, 50, 60, 75, 100 kHz). Four feature selection algorithms (RFECV, PI-RF, PI-LR and GA) were used to select the features, and the results of the variable feature selection are shown in Table 1.

**Table 1.** Screening results of characteristic variables.

| Feature Filtering Methods | Physicochemical Indicators | Number of Features | Features after Filtering | | | | | | |
|---|---|---|---|---|---|---|---|---|---|
| RFECV | SSC | 10 | $25\varepsilon'$ | $30\varepsilon'$ | $60\varepsilon'$ | $100\varepsilon'$ | $20\varepsilon''$ | $25\varepsilon''$ | $50\varepsilon''$ |
| | | | $60\varepsilon''$ | $75\varepsilon''$ | $100\varepsilon''$ | | | | |
| | Acidity | 5 | $30\varepsilon'$ | $20\varepsilon''$ | $25\varepsilon''$ | $40\varepsilon''$ | $50\varepsilon''$ | | |
| | PP | 14 | $10\varepsilon'$ | $12\varepsilon'$ | $15\varepsilon'$ | $20\varepsilon'$ | $30\varepsilon'$ | $40\varepsilon'$ | $50\varepsilon'$ |
| | | | $100\varepsilon'$ | $20\varepsilon''$ | $25\varepsilon''$ | $40\varepsilon''$ | $50\varepsilon''$ | $60\varepsilon''$ | $75\varepsilon''$ |
| PI-FR | SSC | 10 | $4\varepsilon''$ | $0.5\varepsilon'$ | $0.6\varepsilon'$ | $5\varepsilon''$ | $1.2\varepsilon'$ | $0.4\varepsilon'$ | $0.4\varepsilon''$ |
| | | | $1\varepsilon''$ | $3\varepsilon''$ | $6\varepsilon''$ | | | | |
| | Acidity | 11 | $4\varepsilon''$ | $0.5\varepsilon'$ | $7.5\varepsilon''$ | $10\varepsilon''$ | $0.75\varepsilon''$ | $0.6\varepsilon'$ | $0.5\varepsilon''$ |
| | | | $0.2\varepsilon'$ | $5\varepsilon''$ | $0.3\varepsilon''$ | $0.05\varepsilon''$ | | | |
| | PP | 12 | $4\varepsilon''$ | $0.6\varepsilon'$ | $0.5\varepsilon'$ | $6\varepsilon''$ | $3\varepsilon''$ | $0.05\varepsilon'$ | $7.5\varepsilon''$ |
| | | | $0.1\varepsilon''$ | $0.4\varepsilon''$ | $5\varepsilon''$ | $0.05\varepsilon''$ | $2\varepsilon''$ | | |
| PI-LR | SSC | 9 | $10\varepsilon''$ | $75\varepsilon''$ | $40\varepsilon''$ | $12\varepsilon''$ | $60\varepsilon''$ | $30\varepsilon''$ | $25\varepsilon''$ |
| | | | $75\varepsilon'$ | $20\varepsilon''$ | | | | | |
| | Acidity | 9 | $10\varepsilon''$ | $30\varepsilon''$ | $75\varepsilon''$ | $12\varepsilon''$ | $60\varepsilon''$ | $75\varepsilon'$ | $15\varepsilon'$ |
| | | | $25\varepsilon''$ | $40\varepsilon''$ | | | | | |
| | PP | 9 | $10\varepsilon''$ | $75\varepsilon''$ | $12\varepsilon''$ | $30\varepsilon''$ | $25\varepsilon''$ | $40\varepsilon''$ | $75\varepsilon'$ |
| | | | $20\varepsilon''$ | $60\varepsilon''$ | | | | | |
| GA | SSC | 10 | $1\varepsilon'$ | $20\varepsilon'$ | $0.05\varepsilon''$ | $0.15\varepsilon''$ | $0.3\varepsilon''$ | $0.75\varepsilon''$ | $2\varepsilon''$ |
| | | | $3\varepsilon''$ | $6\varepsilon''$ | $40\varepsilon''$ | | | | |
| | Acidity | 7 | $0.05\varepsilon'$ | $1.2\varepsilon'$ | $0.12\varepsilon''$ | $0.75\varepsilon''$ | $1\varepsilon''$ | $2\varepsilon''$ | $50\varepsilon''$ |
| | PP | 9 | $0.05\varepsilon'$ | $1.2\varepsilon'$ | $12\varepsilon'$ | $0.25\varepsilon''$ | $0.75\varepsilon''$ | $1\varepsilon''$ | $5\varepsilon''$ |
| | | | $40\varepsilon''$ | $100\varepsilon''$ | | | | | |

### 3.4. Establishment of Prediction Models and Comparison of Results

Using the full variables (FVs) and variables selected by the RFECV, PI-RF, PI-LR and GA algorithms as inputs, and using passion fruit *SSC*, *Acidity* and *PP* as outputs, XGBoost prediction models (FV-XGBoost, RFECV-XGBoost, PI-RF-XGBoost, PI-LR-XGBoost, GA-XGBoost) and CatBoost prediction models (FV-CatBoost, RFECV-CatBoost, PI-RF-CatBoost, PI-LR-CatBoost, GA-CatBoost) were established. The model parameters were set as shown in Table 2.

**Table 2.** Model parameter settings.

| Modeling Method | Model Parameters | Set Value |
|---|---|---|
| XGBoost | n_estimators | 50 |
| | learning_rate | 0.1 |
| | eval_metric | 'rmse' |
| | max_depth | 10 |
| | objective | 'reg:squarederror' |
| | booster | 'gbtree' |
| CatBoost | iterations | 1000 |
| | learning_rate | 0.01 |
| | eval_metric | 'RMSE' |
| | loss_function | 'RMSE' |
| | depth | 4 |
| | od_type | 'Iter' |

The model results are shown in Table 3. Overall, the prediction performance of the models after feature selection is not significantly different from the full variable regression model. Among them, the PI-RF-XGBoost model demonstrates good performance in predicting *SSC*, achieving a determination coefficient ($R^2p$) of 0.9240 and a root mean square error (RMSEp) of 0.2595 on the prediction set. For predicting *Acidity*, the GA-CatBoost model performs well with an $R^2p$ of 0.9471 and an RMSEp of 0.1237. Similarly, the PI-RF-XGBoost model shows good performance in predicting *PP*, with an $R^2p$ of 0.9092 and an RMSEp of 0.0014. Figure 6 shows the prediction results of the three models. It can be clearly seen that the samples are concentrated near the y = x regression line, and the prediction results are better.

**Table 3.** XGBoost and CatBoost modeling results. The results are presented as the mean of each fold.

| Physicochemical Indicators | Feature Filtering Methods | XGBoost | | | | CatBoost | | | |
|---|---|---|---|---|---|---|---|---|---|
| | | Calibration Set | | Prediction Set | | Calibration Set | | Prediction Set | |
| | | $R^2c$ | RMSEc | $R^2p$ | RMSEp | $R^2c$ | RMSEc | $R^2p$ | RMSEp |
| SSC | FV | 0.9986 | 0.2175 | 0.8992 | 0.2175 | 0.9918 | 0.3605 | 0.9116 | 0.3605 |
| | RFECV | 0.9940 | 0.3529 | 0.8950 | 0.3529 | 0.9729 | 0.6611 | 0.8682 | 0.6611 |
| | PI-RF | 0.9975 | 0.2595 | 0.9240 | 0.2595 | 0.9822 | 0.5269 | 0.9152 | 0.5269 |
| | PI-LR | 0.9958 | 0.3120 | 0.8958 | 0.3120 | 0.9702 | 0.6879 | 0.8743 | 0.6879 |
| | GA | 0.9980 | 0.2411 | 0.9040 | 0.2411 | 0.9858 | 0.4746 | 0.9031 | 0.4746 |
| Acidity | FV | 0.9812 | 0.2410 | 0.9171 | 0.2410 | 0.9973 | 0.0875 | 0.9397 | 0.0875 |
| | RFECV | 0.9649 | 0.3256 | 0.8960 | 0.3256 | 0.9845 | 0.2137 | 0.8786 | 0.2137 |
| | PI-RF | 0.9757 | 0.2712 | 0.9233 | 0.2712 | 0.9949 | 0.1211 | 0.9382 | 0.1211 |
| | PI-LR | 0.9687 | 0.3057 | 0.9069 | 0.3057 | 0.9895 | 0.1748 | 0.9226 | 0.1748 |
| | GA | 0.9760 | 0.2680 | 0.9257 | 0.2680 | 0.9947 | 0.1237 | 0.9471 | 0.1237 |
| PP | FV | 0.9998 | 0.0012 | 0.9016 | 0.0012 | 0.9577 | 0.0117 | 0.8740 | 0.0195 |
| | RFECV | 0.9986 | 0.0024 | 0.9016 | 0.0012 | 0.9188 | 0.0163 | 0.8740 | 0.0195 |
| | PI-RF | 0.9997 | 0.0014 | 0.9092 | 0.0014 | 0.9483 | 0.0129 | 0.8708 | 0.0199 |
| | PI-LR | 0.9982 | 0.0027 | 0.8859 | 0.0027 | 0.9033 | 0.0176 | 0.8056 | 0.0244 |
| | GA | 0.9996 | 0.0015 | 0.9030 | 0.0015 | 0.9464 | 0.0132 | 0.8748 | 0.0194 |

Furthermore, for the algorithms that performed well, their input features are mainly concentrated in the frequency range of 0.05–6 kHz, which is consistent with the previous results. From the perspective of the complexity of the regression models, the RFECV, PI-RF, PI-LR and GA algorithms have significantly simplified the computational complexity of the models. The number of variables has been reduced by 85.78%, 83.82%, 86.77% and 87.26%, respectively. Moreover, by pruning the models and reducing the number of iterations, the model size has been further reduced. The results indicate that the PI-RF-XGBoost,

GA-CatBoost and PI-RF-XGBoost models can provide good predictions of passion fruit's *SSC*, *Acidity* and *PP* within 1 ms.

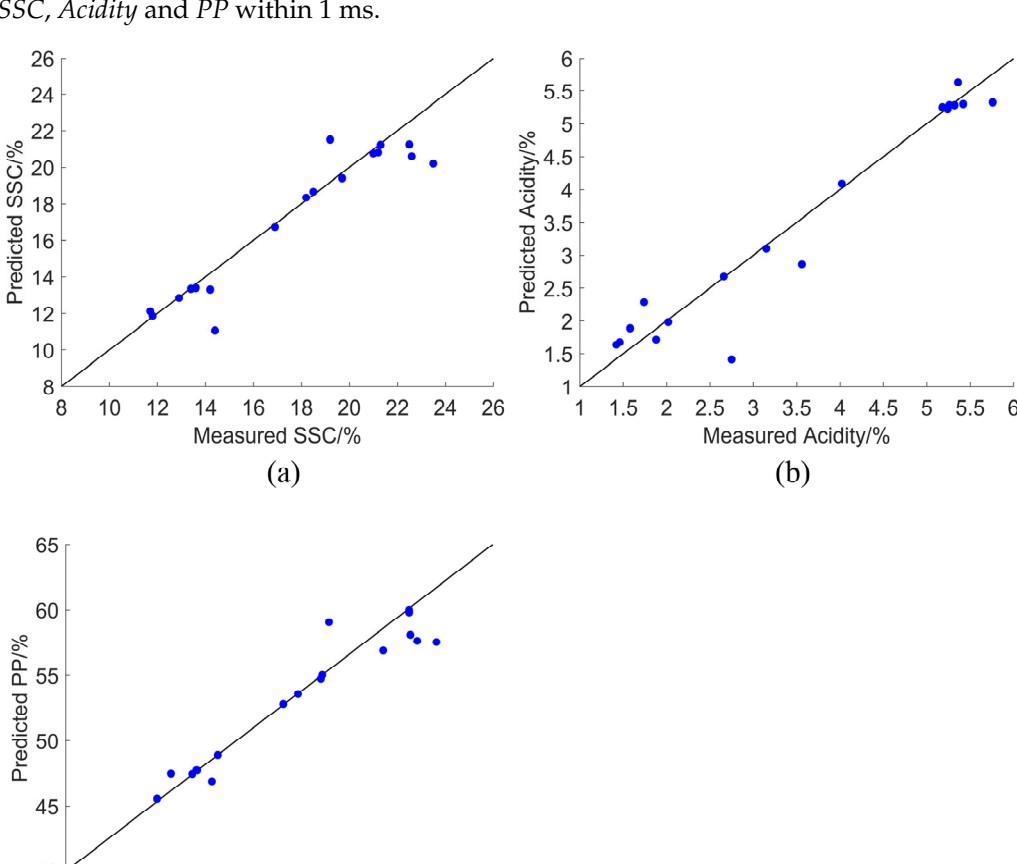

**Figure 6.** Comparison between measured and predicted results of fruit physicochemical parameters: (**a**) *SSC*; (**b**) *Acidity*; (**c**) *PP*.

The differences in $R^2$ values between the calibration and prediction sets for the three predictive models are 0.0735, 0.0476 and 0.0905, respectively. Overall, the robustness of the three models is good, and there is no issue of overfitting leading to inadequate generalization ability. However, in general, the prediction performance for *SSC* and *Acidity* is better than that for *PP*. This difference may be related to the reason for the dielectric behavior, which is mainly caused by free water propagation and ion migration within the passion fruit [12].

**4. Conclusions**

In this study, a detection platform was established using the LCR digital bridge tester, and the dielectric properties of 192 passion fruit samples were accurately measured at 34 different frequencies ranging from 0.05 to 100 kHz. After removing outliers and conducting feature selection, prediction models based on dielectric properties were established for quality indicators of passion fruit. The main conclusions are as follows:

1. There are differences in the $\varepsilon'$ and $\varepsilon''$ parameters of passion fruit with different maturities. Moreover, significant correlations exist between the physicochemical indicators of passion fruit and the $\varepsilon'$ and $\varepsilon''$ parameters at frequencies ranging from 0.05 to 10 kHz. The correlation coefficients range between 0.6 and 0.85.

2. The Local Outlier Factor (LOF) algorithm is used to filter out 12 outlier samples. Recursive Feature Elimination with Cross-Validation (RFECV), Permutation Importance based on Random Forest Regression (PI-RF), Permutation Importance based on Linear

Regression (PI-LR) and Genetic Algorithm (GA) are employed to select 68 dielectric parameter features. The number of variables has been reduced by 85.78%, 83.82%, 86.77% and 87.26%, respectively.

3. The XGBoost and CatBoost algorithms are used to establish a quality prediction model for passion fruit and achieved good detection results. The determination coefficients ($R^2$) for predicting the physicochemical parameters (*SSC*, *Acidity*, and *PP*) are 0.9240, 0.9471 and 0.9092, respectively.

The results of this study prove the effective application of the feature screening method and the XGBoost and CatBoost algorithms in the field of agricultural science and tech-nology, especially the prediction of passion fruit quality indicators based on dielectric properties. In future work, it is possible to predict the quality characteristics of postharvest fruit, such as maturity, moisture content, hardness and vitamin C content, based on their dielectric properties. The growth status of fruit can be estimated through physicochemical indicators. By applying appropriate postharvest treatments to fruit of different maturity, the quality of the fruit can be ensured during the shelf life, thus enhancing their economic value.

However, the study still has the following limitations: the experimental samples were of a single variety and from a limited geographical area, the equipment cost was high and the operation was complex. In order to better apply this technology to agricultural production practice, future studies could conduct: experiments on fruit of multiple varieties and from different geographical locations to identify commonalities and differences among them; fixed-frequency dielectric detection sensors could also be used in the future to create a more convenient and efficient fruit quality detection device.

**Author Contributions:** F.L.: Conceptualization, Investigation, Methodology, Writing—original draft. D.C.: Investigation, Methodology, Writing—review & editing. C.L.: Investigation, Methodology. J.H.: Funding acquisition, Resources, Supervision, Writing—review & editing. All authors have read and agreed to the published version of the manuscript.

**Funding:** This research was funded by Fujian Province Agricultural Key Core Technology Research Project (KLY23416XA).

**Institutional Review Board Statement:** Not applicable.

**Informed Consent Statement:** Not applicable.

**Data Availability Statement:** The data presented in this study are available on request from the corresponding author.

**Conflicts of Interest:** The authors declare no conflicts of interest.

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
