# Peer review of "Non-Destructive Detection of Golden Passion Fruit Quality Based on Dielectric Characteristics"

_applsci, doi:10.3390/app14052200_

Round 1
Reviewer 1 Report
Comments and Suggestions for Authors
I spent quite a bit of time reviewing this paper. However, I decided to express my surprise. In order to review it, I need to know what type of scientific paper is it? Article, Review, Communication, etc.?
It is not mentioned in the PDF, and on the platform in the Review Report Form section it is written "Type Essay", but with certainty, this scientific paper is not an essay. Only depending on this, and necessarily, I can review this work. Without this information I cannot appreciate it.
If the authors still want to classify it as an essay, then I consider that this work cannot be published as an essay. But, if it is an article, then I think it is a good one, which could be published, although it requires some additions and changes.
I am waiting for this information, after which I will return with the review of the work.

Author Response
Dear reviewer,
I'm very sorry for my mistake in choosing the paper type and causing you bad confusion and a waste of time. There is no doubt that the type of this manuscript is article. I hope you can give me more suggestions on my manuscript, and I will revise it seriously. Thank you very much for your opinion.
Sincerely
Fan Lin

Reviewer 2 Report
Comments and Suggestions for Authors
Work is good but needs some corrections

Comments on the Quality of English Languageok
Author Response
Dear reviewer,
Our revised response has been submitted, please review it again.
If there are any problems, we will actively modify them.
Sincerely
FanLin

Reviewer 3 Report
Comments and Suggestions for Authors
The development of novel non-destructive and fast methods to monitor the physicochemical state and control the quality of fruits and vegetables is highly relevant not only for science but also for practical applications.
The measurement of electrical characteristics, such as dielectric behavior, of the material can address these challenges.
In the manuscript applsci-2876623 , the applicability of dielectric parameters (dielectric constant and dielectric loss factor) for monitoring the ripeness and physicochemical quality of golden passion fruit was investigated using a low-frequency range (0.05 -100 kHz) for measurements. Among the physicochemical quality indicators, the soluble solid content, pulp content, and acidity were determined. The applied algorithms and screening methods (LOF, RFECV, PI-RF/LR, XGBoost, Cat Boost, etc.) are appropriate.
The manuscript has a logical structure. The introduction section summarizes the background and relevance of the study well. The novelties and specific aims of the research are clearly defined.
The applied measurement and calculation methods are described clearly.
The manuscript contains novel, interesting, and valuable results that are presented clearly.
Comments and suggestions:
In my opinion, the conclusion is too general. Please rephrase it to provide information related to the determined dielectric parameters and control parameters (physicochemical parameters).
Please provide details on how the temperature was controlled during the dielectric measurements (dielectric parameters can be influenced by temperature).
Please explain how the frequency range for dielectric measurements was selected.
Please improve the visibility of Figures 3-4 (mainly axis titles, scales, and units).
Please correct the title of Table 2 (line 314).
The authors explain the dielectric behavior superficially in lines 348-350. Please briefly discuss the biochemical changes of the fruit during ripening as well as their possible effects on the dielectric parameters.
Author Response

(The authors gave the same response as above.)

Round 2
Reviewer 1 Report
Comments and Suggestions for Authors
The article is well-designed and presents valuable and relevant research for the field. I congratulate the authors for it!
After the first review that has already been done, some additions would be necessary, to raise the value of the article and increase the chances of citation in the future, with contributions to future research in the field.
These additions are aimed at specifying future research directions. The authors mention in the Conclusions section, the fact that "However, this study still has some shortcomings. There are few studies on the microscopic changes and physiological and biochemical research of fruit ripening. And the test subjects only considered golden passion fruit of a single variety in a single orchard. This requires continuous research and efforts by scholars. In order to better apply this technology to agricultural production practices, fixed-frequency dielectric detection sensors can be used in the future to create a more convenient and efficient fruit quality detection device. The selection of these specific frequencies can be guided by the findings of this study. This research provides valuable insights for the non-destructive testing and grading of passion fruit quality. ". However, the authors do not specify concretely, to the point, what would be the recommended future research directions, and how they could add value to the current level of knowledge.
Author Response
Dear reviewer,
We believe that this revision has clarified the research value of the article and the future research direction. And the structure of the article is clearer. If you has any other questions, we will actively cooperate and correct them. Thank you!
Sincerely,
Fan Lin
